# The Effect of Powder Composition on the Microstructure and Corrosion Resistance of Laser Cladding 60NiTi Alloy Coatings on SS 316L

**Zexu Du, Zhengfei Hu \*, Yuqiang Feng and Fan Mo**

Shanghai Key Laboratory for R&D and Application of Metallic Functional Materials, School of Materials Science and Engineering, Tongji University, Shanghai 201804, China; 1832875@tongji.edu.cn (Z.D.); yuqiang_feng@tongji.edu.cn (Y.F.); mofan61@163.com (F.M.)
**\*** Correspondence: huzhengf@tongji.edu.cn; Tel.: +86-21-6958-5265

**Abstract:** Two kinds of 60NiTi powders were prepared by pure Ni mixed with Ti powders, and 55NiTi alloy powder with pure Ni powder and both the powders were fully mixed by alcohol ball milling. Two kinds of coatings (denoted as 60Ni-40Ti and 55NiTi-5Ni) were prepared on a 316L stainless steel substrate by laser cladding. The microstructure, microhardness and electrochemical behavior of the prepared coatings were investigated extensively. The results show that 55NiTi-5Ni has a typical dendritic eutectic structure, but 60Ni-40Ti tends to form a eutectic network structure. The main phases in both coatings are (Ni, Fe)Ti and (Ni, Fe)$_3$Ti; however, the (Ni, Fe)Ti phase is dominant in 55NiTi-5Ni, but the (Ni, Fe)$_3$Ti phase is more prevalent in 60Ni-40Ti. The microhardness was significantly improved with the 316L stainless steel substrate, and the microhardness of 55NiTi-5Ni is slightly higher than 60Ni-40Ti. The corrosion resistance of the two coatings in 3.5 *wt*% NaCl solution also leads to significant improvements compared with the substrate, and the corrosion resistance of 55NiTi-5Ni was also increased. These different behaviors and characteristics might be related to the different microstructures. Uniform and fine eutectic structure in 55NiTi-5Ni coating lead to better performance, which is also conducive to the formation of the dense oxide film to improve corrosion resistance.

**Keywords:** laser cladding; nickel-titanium alloy coating; microstructure; microhardness; electrochemical behavior

## 1. Introduction

NiTi alloy was first developed in the Naval Ordnance Laboratory by William J. Buenler and his colleagues [1] in the late 1950s and has been increasingly used in biomedical, automotive, aerospace, telecommunication, navigation and other fields [2,3]. 55NiTi (55 *wt*% Ni (50 *at*% Ni)) alloy with near equal atomic ratio has shape memory effect and hyperelasticity. It is a typical shape memory alloy and has attracted extensive attention from researchers [4]. 60NiTi (60 *wt*% Ni (55 *at*% Ni)) alloy was also discovered during the same period, and recent studies have shown that 60NiTi has the advantages of high hardness, strong corrosion resistance, high wear resistance, low elastic modulus and so on. Khanlari [5] et al. showed that the hardness of 56~62 HRC and friction coefficient of 0.06 of 60NiTi alloy were similar to those of 440C bearing steel (Fe-0.95C-16Cr-0.75Mo, *wt*.%), while Young's modulus of 95 GPa was only half of that of 440C bearing steel. Qin [6] et al. showed that the corrosion rate of 60NiTi in 3.5 *wt*% NaCl was about 1/1000 of that of 316 stainless steel (Fe-0.03C-17Cr-12Ni-2.5Mo, *wt*%).

However, 60NiTi received little attention at that time because of its brittleness, which is hard to machine and difficult to work with [7]. Recently, the application of more advanced manufacturing and processing methods is emerging, and the preparation of 60NiTi alloy has been studied gradually. Casting [4] and powder metallurgy [8] are currently the

main methods for 60NiTi alloy manufacturing. Nevertheless, 60NiTi is not a cost-effective material due to its high price and complicated preparation process. With the development of surface modification technology, 60NiTi alloy coating has become the focus of research. Based on the statistical data, more than 80% of machine failures are caused by abrasion on the surface of the component [9]. Accordingly, it can be expected that the machine life will be remarkably elongated by preparing a coating of the 60NiTi alloy. Zhang [10] et al. prepared 60NiTi alloy coating by magnetron sputtering, which obviates the need for shaping 60NiTi alloy, and its hardness and corrosion resistance were significantly improved compared with 316 stainless steel substrate.

As a surface modification technology, laser cladding is a method of material deposition by which a powder material is melted and consolidated by the use of a laser in order to coat substrate [11]. Laser cladding has the advantages of low heat input, fast forming speed and high forming accuracy [12]. This technique was initially developed for surface treatment with the aim to produce surface coatings (e.g., wear-resistant, corrosion-resistant or biocompatible coatings) and repair high-value components (such as tools, turbine blades, shafts and valves) [13]. However, a drawback of laser cladding technology is that the corrosion resistance of cladding is not as effective as the same bulk materials [14]. It can be improved by selecting materials with strong corrosion resistance and optimizing the surface quality [15] of the coating.

This provides a suitable processing method for preparing NiTi alloy coatings, and some research on laser cladding nickel–titanium alloy has been carried out. Liu [16] et al. studied a gradient Ni-Ti coating that was laser cladded on the surface of the TA2 titanium alloy (Ti-0.1C-0.3Fe-0.25O, $wt$%) substrate, and the microstructure and oxidation behavior of the laser-cladded coating were investigated experimentally. Fu [17] et al. investigated microstructure and properties of laser-cladded NiTi and NiTiCu coatings on TA2 titanium alloy substrate to reveal the effect of Cu addition to NiTi coatings. Current studies are more or less focused on Ti-rich substrates because of the low precipitation of the Ti-rich part of the elemental diffusion between the substrate and the coating based on the Ni-Ti binary phase diagram. However, in most of the abrasion resistance engineering applications, the partial failure of the workpiece is mainly the steel material. Scholars have also carried out research on the laser cladding NiTi alloy coating on steel. Norhafzan [18] et al. presented laser cladding of 55NiTi powder on an H13 tool steel (Fe-0.32C-4.75Cr-0.20Mn-0.80Si-0.80V, $wt$%) surface for enhancement of surface properties.

316L stainless steel is a widely used engineering alloy in liquid-handling systems and hydraulic machinery because of its excellent corrosion resistance, good processability and relatively low cost. However, its resistance to cavitation erosion is low, mainly because of low hardness (around 200 HV), and its use is thus limited to mildly cavitating environments [15,19]. Laser cladding of 316L using NiTi alloy coating is a promising application. Chiu [20] et al. showed the excellent corrosion resistance of laser cladding a 55NiTi alloy strip on 316L stainless steel. The research focus was mainly on the preparation process and the characterization of material properties, without an in-depth study of the microstructure and its combination with performance.

Among the previous studies on NiTi alloy cladding coating, the main study material was 55NiTi alloy; in other words, few studies have been based on laser cladding of 60NiTi alloy, not only because of the late start of 60NiTi alloy research but also due to its high cost and brittleness, which makes it difficult to prepare laser-processed spherical powder or wire [21].

In this study, on the basis of not directly preparing 60NiTi alloy powder, two kinds of 60 $wt$.% NiTi powders were prepared and ball-milled. They were coated on 316L stainless steel using laser cladding technology. The effect of different powder compositions of 60NiTi on the microstructures and properties was investigated carefully.

## 2. Experimental Methods

In this study, the substrate was SS 316L with a specification of 100 mm × 100 mm × 10 mm. The raw powder materials were commercial pure Ni powder, pure Ti powder and 55NiTi alloy powder with a particle size of 53–106 μm. The nominal composition of powders is shown in Table 1. Two kinds of 60 *wt*% Ni-mixed powders were prepared using pure Ni and Ti powders and 55NiTi alloy powder with pure Ni powder. The alloy powders were mechanically blended with alcohol using ball mill pots and milled with a planetary ball for 480 min with a constant rotation speed of 224 rpm. The SEM (Zeiss Sigma 500, Zeiss, Oberkochen, Germany) images and XRD (D8 ADVANCE, BRUKER AXS, Karlsruhe, Germany) patterns of the powder after vacuum drying are shown in Figure 1 [22].

**Table 1.** The nominal composition of the powders (*wt*%).

| Powder | Ni | Ti | Co | Na | Cu | Si | Fe | C | O |
|---|---|---|---|---|---|---|---|---|---|
| Pure Ni | Bal. | - | 0.020 | <0.005 | 0.001 | 0.003 | 0.003 | 0.002 | 0.006 |
| Pure Ti | - | Bal. | - | - | - | - | 0.016 | 0.006 | 0.058 |
| Alloy 55NiTi | 56.46 | Bal. | - | <0.005 | <0.005 | - | <0.005 | 0.003 | 0.037 |

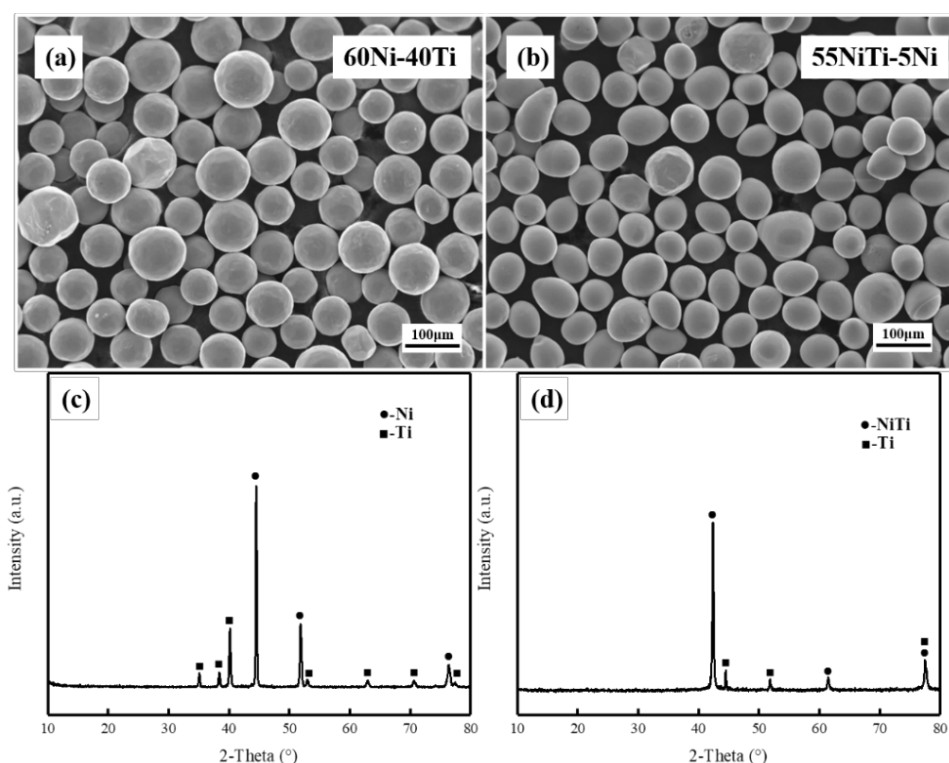

**Figure 1.** Scanning electron microscopy (SEM) images and X-ray diffraction (XRD) patterns of mixed powders: (**a**,**c**) 60Ni-40Ti; (**b**,**d**) 55NiTi-5Ni.

In this work, the synchronous powder feeding method was used for laser cladding. The coatings were fabricated by a high-power fiber laser processing machine (IPG YLS-, IPG Photonics, Oxford, MA, USA) with a wavelength of 1070 nm. The substrate was pre-heated to 200 °C to reduce the temperature gradient [17]. The control variable method was used to design three strong experimental parameters of laser power, feeding speed and scanning speed. The results show that the best quality parameters of the cladding coating operation were a laser power of 2000 W, scanning speed of 4 mm/sand powder feeding speed of 50 rpm. During the laser cladding process, the high purity argon was injected with a flow rate of 5 L/min as shielding gas and a flow rate of 8 L/min as powder feeding gas.

Observation specimens were cut perpendicular to the laser scanning direction. The surfaces of the specimens were mechanically ground with sandpaper (200#, 600#, 1000#, 1500#, 2000#, 2500#) and polished with $SiO_2$ powders with the size of 1 μm, then cleaned in deionized water, rinsed in ethanol and dried. After that, they were etched in a metallographic etchant ($HF/HNO_3/H_2O = 1:4:5$). The phase composition of the coatings was analyzed by X-ray diffraction at a scanning speed of $10°/min$, ranging from $10°$ to $80°$, with Cu Kα at 40 kV and 40 mA. Optical microscopy (OM, OLYMPUS GX5, OLYMPUS, Tokyo, Japan) was utilized to observe the microstructures. Field emission scanning electron microscopy with an energy spectrum analyzer (EDS, OXFORD Ultim Extreme, OXFORD, OXFORD, England) was used to analyze the microstructure and elementary composition of the coatings.

According to the ASTM standard E92-82 [23], the microhardness of the coatings was measured with a Vickers hardness tester (HVS-1000A, HuaYin Testing Machine Technology Co., Ltd, Laizhou, China) with a load of 1.961 N and a duration of 15 s. The hardness test position was from the top of the coating to the substrate at 0.2 mm intervals. At each test position, two more positions were tested at an interval of 0.1 mm from the horizontal direction, and the average value of the three test values was calculated to avoid accidental results [24]. The samples for the electrochemical experiment were sealed with epoxy resin and polished. The test was carried out on an electrochemical workstation (GAMRY Reference 600+, GAMRY, Warminster, PA, USA) with 3.5 *wt*% NaCl solution at room temperature. A Hg/HgO electrode was used as the reference electrode, and a platinum electrode was selected as the auxiliary electrode. The open-circuit potential (OCP) of anodes was tested for 2 h to reach a stable potential of the tested anodes. The potentiodynamic polarization (PD) test was carried out from −0.5 (vs. open-circuit potential) to 1.5 V at a scanning rate of 0.1 mV/s. Electrochemical impedance spectra (EIS) were also measured to evaluate the electrochemical behavior at the frequency ranging from 100,000 to 0.01 Hz at the amplitude of 5 mV.

## 3. Results and Discussion

### 3.1. Phase Structures

Figure 2 depicts the XRD patterns of the coatings. There are mainly three different types of diffraction peaks, which are (Ni, Fe)Ti, (Ni, Fe)$_3$Ti and Ni$_4$Ti$_3$. In addition, a small amount of $TiO_2$ and $Fe_2Ti$ also exist in the coatings [25,26]. XRD patterns show that the number of major phase characteristic peaks of the coatings is similar, indicating that the phase composition in the two coatings is similar.

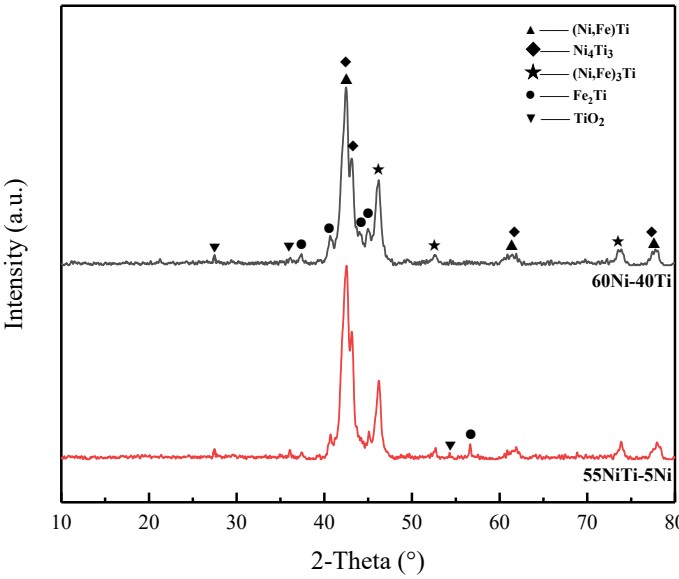

**Figure 2.** XRD patterns of coatings.

### 3.2. Microstructural and Compositional Analysis

Figure 3 illustrates the cross-sectional OM images of the two coatings, and there are no obvious cracks or pores in the cross-section of the coatings. A clear boundary line can be found between the coating and the substrate, which indicates good metallurgical bonding. The laser cladding layer is mainly composed of the cladding zone (CZ), bonding zone (BZ), and heat-affected zone (HAZ) [24]. Dilution rate $\eta$ ($\eta = h/(H + h)$) is an important parameter in the quality characterization of a laser-cladding coating, where H is the melting height and h is the melting depth. According to the dilution rate of two coatings given in Table 2, it can be seen that the dilution rate of 60Ni-40Ti (0.54) is higher than that of 55NiTi-5Ni (0.36). A higher dilution rate means that more Fe, Cr and other substrate elements are diffused into the cladding coating, which is detrimental to the performance of the coating [27]. To facilitate the expression of subsequent analysis, as shown in Figure 3, the top, middle, and bottom of the cladding zone are marked as Top, Mid and Bot, respectively.

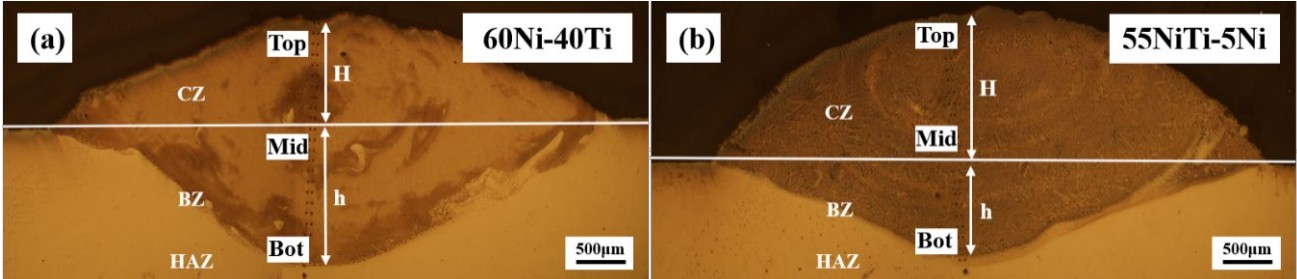

**Figure 3.** Optical microscopy (OM) images of coatings: (**a**) 60Ni-40Ti; (**b**) 55NiTi-5Ni.

**Table 2.** Dilution rates of coatings.

| Sample | H (mm) | H (mm) | η |
|---|---|---|---|
| 60Ni-40Ti | 1.645 | 1.393 | 0.54 |
| 55NiTi-5Ni | 0.99 | 1.745 | 0.36 |

Figure 4 shows the SEM images near the bonding zone. The width of the bonding zone of the 60Ni-40Ti (26.89 μm) is smaller than that of 55NiTi-5Ni (38.56 μm), which means the transition of elements in the interface bonding zone of 55NiTi-5Ni is relatively slow. The wider transition zone indicates that the metallurgical bonding between the coating and the substrate is stronger, and the cladding quality is better [28].

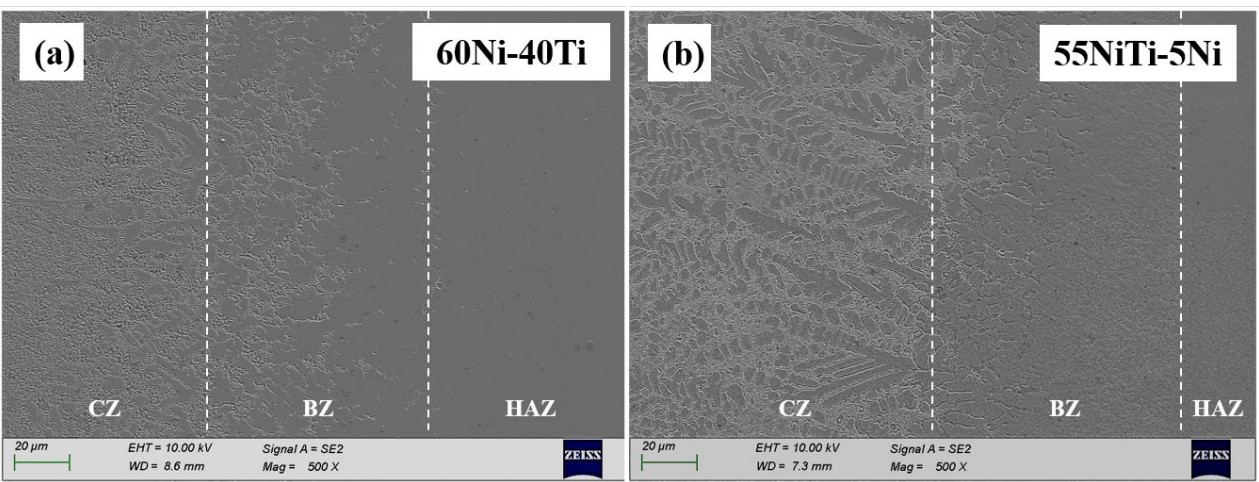

**Figure 4.** SEM images of the bonding zone of coatings: (**a**) 60Ni-40Ti; (**b**) 55NiTi-5Ni.

The Top, Mid and Bot areas of the two coatings were observed by SEM, and the chemical composition was analyzed with EDS. It was found that the microstructure of the two coatings at different areas was almost similar. The Top area of the coatings was selected for microstructure investigation and acted as the working face of the electrochemical behavior test. As shown in Figure 5a–c, similar to the laser cladding mixed pure powders, 60Ni-40Ti tends to form a eutectic network structure. However, 60Ni-40Ti coating grows with a coarse network structure and fine network structure alternately, which is different from the laser cladding of nickel–titanium pure mixed powders [17,27]. As exhibited in Figure 5d–f, 55NiTi-5Ni shows a typical dendritic eutectic structure, which is composed of coarse dendritic and interdendritic eutectic structures. In contrast to the coating of laser cladding 55NiTi alloy powder, the eutectic structure in 55NiTi-5Ni is finer. In addition, black petal-like precipitates were observed in both coatings. The size of the precipitates was about 5 μm with different shapes, and they mainly grew on the grain boundaries. Figure 5g–i shows the microstructure of the two coatings at a 2000× where the overall grain size of 55NiTi-5Ni coating is much finer than that of 60Ni-40Ti coating. It is due to the dense dendritic structure of 55NiTi-5Ni and the fine eutectic structure between them.

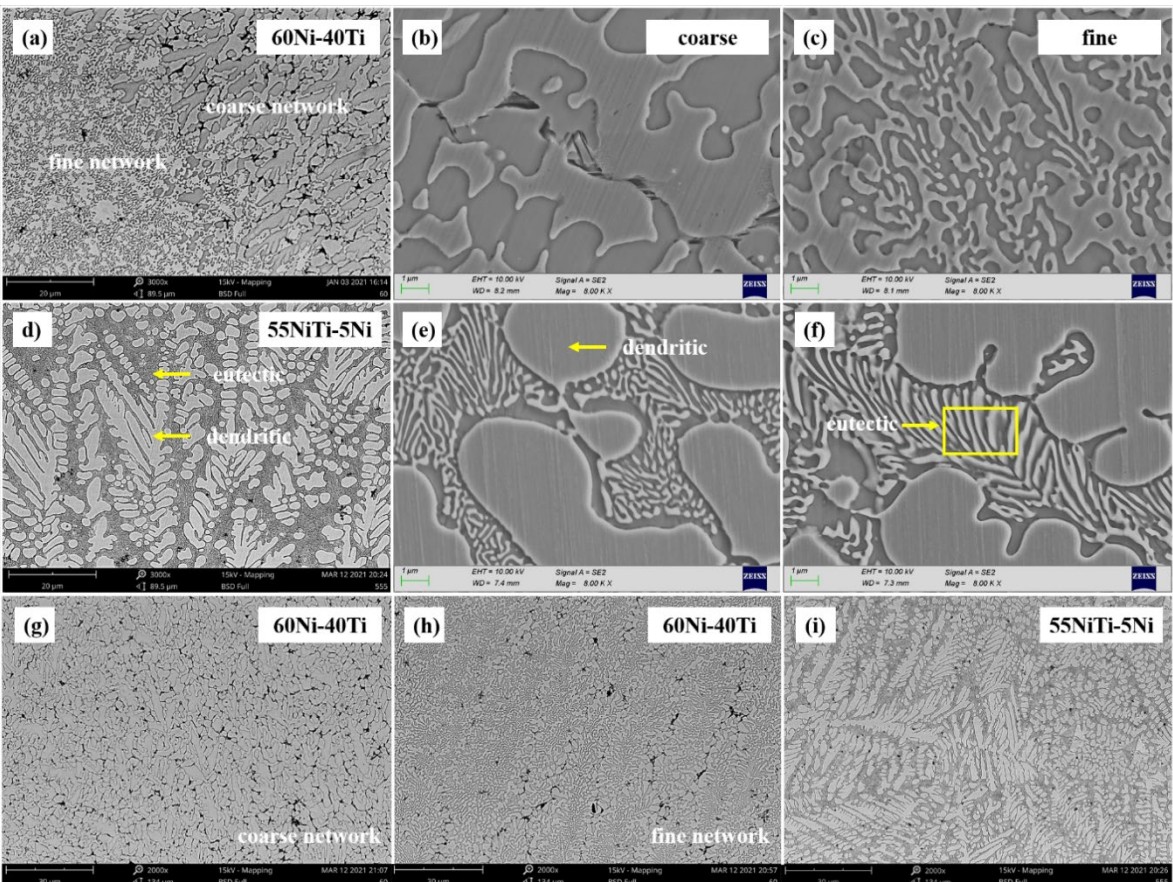

**Figure 5.** Backscattered electron (BSE) and SEM images of the microstructure of coatings: (**a**–**c**) 60Ni-40Ti; (**b**–**f**) 55NiTi-5Ni; BSE images of coatings at low magnifications to compare grain size: (**g**–**i**).

In order to more accurately calibrate the chemical composition of each zone by EDS, SEM images with a magnification of × were collected at the Top area of the two coatings. As shown in Figure 6a–d, there are two main types of phase zones in the organizational structures of 60Ni-40Ti: the net-like zone and the internet-like zone marked A and B, respectively. There are three main phase zones of 55NiTi-5Ni: the dendritic zone, the interdendritic lamellar zone and the interlamellar zone marked C, D and E, respectively. In

addition, petal-like precipitates in both coatings were marked F. The elemental composition of zones A–E is shown in Table 3.

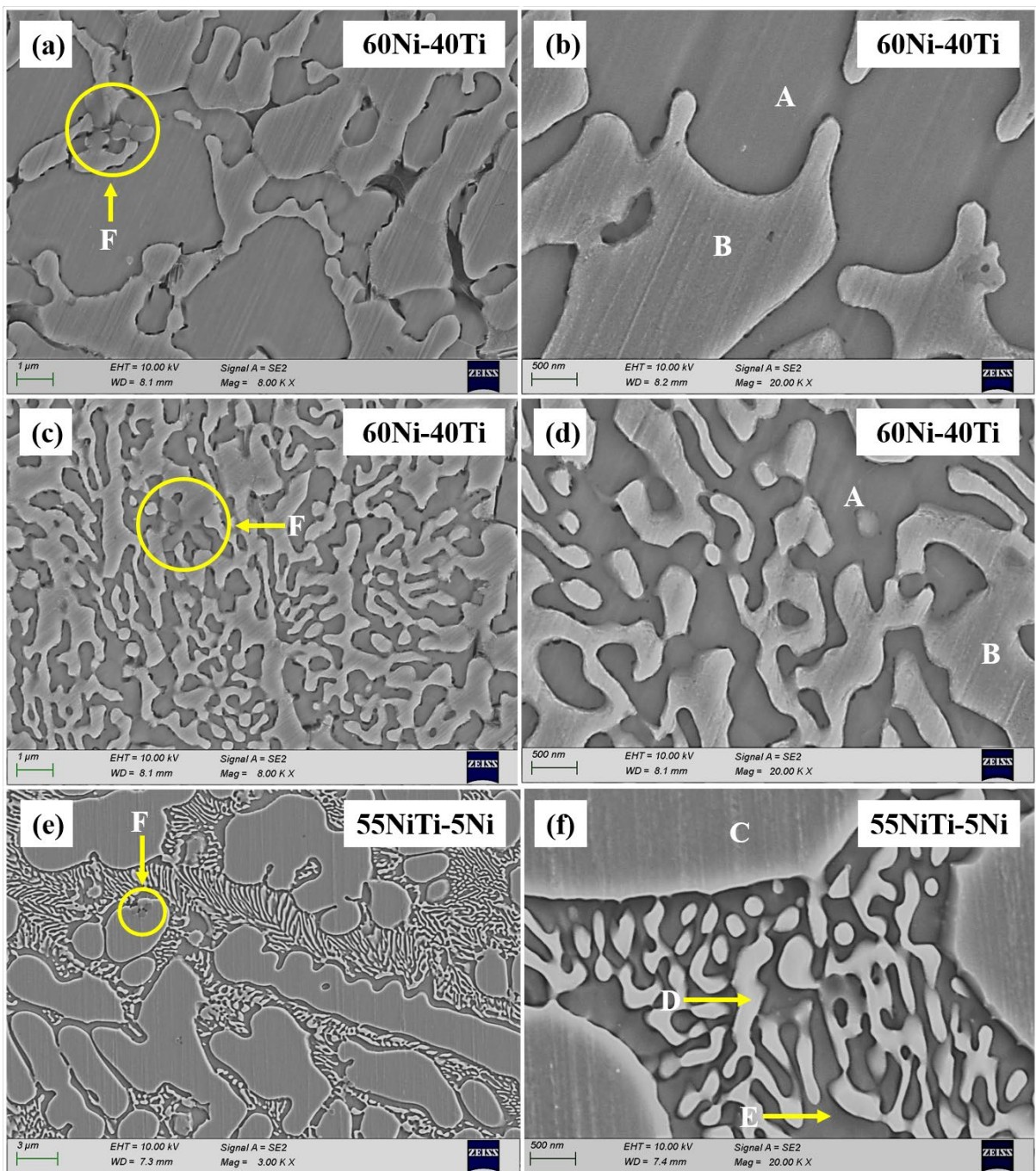

**Figure 6.** SEM images of the Top area: (**a**–**d**) 60Ni-40Ti; (**e**–**f**) 55NiTi-5Ni.

**Table 3.** The result of the elemental composition of zones A–E in Figure 6.

| Element | 60Ni-40Ti (Coarse) | | 60Ni-40Ti (Fine) | | 55NiTi-5Ni | | |
|---|---|---|---|---|---|---|---|
| | **A** | **B** | **A** | **B** | **C** | **D** | **E** |
| Ni (At%) | 44.7 | 30.2 | 40.12 | 28.03 | 35.31 | 38.96 | 64.89 |
| Ti (At%) | 14 | 27.13 | 13.76 | 26.31 | 31.62 | 29.37 | 22.8 |
| Fe (At%) | 32.85 | 35.69 | 37.35 | 36.39 | 26.16 | 27.16 | 9.62 |
| Cr (At%) | 8.45 | 7.99 | 8.77 | 9.29 | 6.91 | 8.51 | 2.69 |

Among the main elements detected in the two coatings, the content of Cr was the lowest. As for solid solution elements, Cr has the same influence as Fe on the phase formation of the coatings, so only Ni, Ti and Fe were considered in phase deduction [29].

In the 60Ni-40Ti coating, the proportion of Ti in zone A is lower than that in zone B. Based on the XRD and EDS results, it can be deduced that zone A is (Ni, Fe)$_3$Ti, and zone B is (Ni, Fe)Ti. The Ni-Ti binary phase diagram shows that the reaction of Ni and Ti in the nickel-rich region generates Ni$_3$Ti and NiTi with the decrease in temperature [30]. Under the action of a high-energy laser, the cladding powder and the surface layer of SS 316L substrate melt, which can accelerate the convection in the formed molten pool and element diffusion between coating and substrate can occur. Fe and Cr atoms diffused from the substrate exist in the Ni-Ti phase in the form of solid solution due to its similar electronegativity to Ni [31].

It can be seen that the Fe element is more soluble in the NiTi phase because the Fe element can more easily replace the Ni element in the B2 lattice. Bozzolo [32] et al. studied the atomic substitution position of the third alloying element in NiTi-X alloy by atomic model and observed the same results.

In the 55NiTi-5Ni coating, the composition of zone C is similar to that of zone D, whereas zone E is rich in Ni. On the basis of the XRD results, zones C and D are (Ni, Fe) Ti, and zone E is (Ni, Fe)$_3$Ti. The 55NiTi-5Ni mixed powder is mainly composed of alloy powder, which is mainly composed of B2 austenitic NiTi phase. It is melted together with SS 316L substrate to form a molten pool, and Fe and Cr exist in the NiTi phase in the form of a solid solution. Then Ni$_3$Ti and NiTi are formed during the rapid solidification process. Unlike 60Ni-40Ti, only a few Fe and Cr atoms can form a solution in the Ni$_3$Ti matrix. In addition to the difference in the solid solubility of Fe atom in the NiTi phase, this is because the reaction to form Ni$_3$Ti is carried out after the formation of NiTi, and the solid solution of Fe and Cr atoms is reduced in the rapid condensation process. In addition, the proportions of the main phase of the two coatings are different. The volume fraction of the (Ni, Fe)Ti phase in the two coatings was estimated and quantified using Image-Pro Plus at low magnification according to SEM images. The calculated proportion was 41.91% and 64.04% for 60Ni-40Ti and 55NiTi-5Ni, respectively.

EDS scanning results of petal-like particles of zone F (shown in Figure 6) in the two coatings shown in Figure 7 indicate that they are of the Ti-rich phase, and no Ni, Fe or Cr elements were present in these particles. It is believed that they are Ti oxides, which is also consistent with the detection of a small amount of Ti-oxide in the XRD patterns. The Ni$_4$Ti$_3$ phase was also detected in the XRD pattern, but no Ni$_4$Ti$_3$ phase was found in the above-mentioned microstructure observation. This is due to the small size of the Ni$_4$Ti$_3$ phase and its co-lattice with the NiTi phase [8], which is also the key factor for the existence of a high hardness of 60NiTi [33]. Moreover, Fe$_2$Ti shown in XRD patterns was not found in the above-mentioned microstructure observation because of its lower content, which may be formed during the solidification stage based on the Fe-Ni-Ti ternary phase diagram [34].

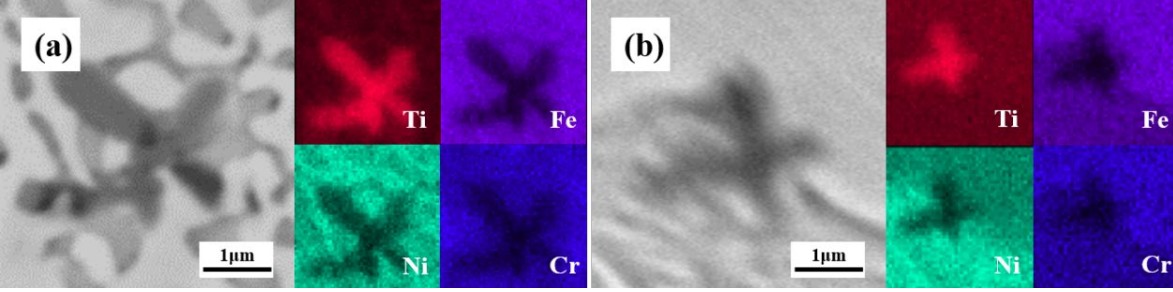

**Figure 7.** SEM images and EDS map scanning results of the petal-like precipitates: (**a**) 60Ni-40Ti; (**b**) 55NiTi-5Ni.

### 3.3. Microhardness

Figure 8a shows the microhardness test results of the coatings. It is apparent that the microhardness curves of both coatings are divided into three areas, the cladding zone, bonding zone, and heat-affected zone. Moreover, the microhardness of both coatings is significantly higher than that of the substrate and the as-cast 60NiTi alloy (400 HV). The higher hardness may be attributed to the agglomeration of nickel–titanium particles [18] and the grain refinement caused by rapid cooling [22].

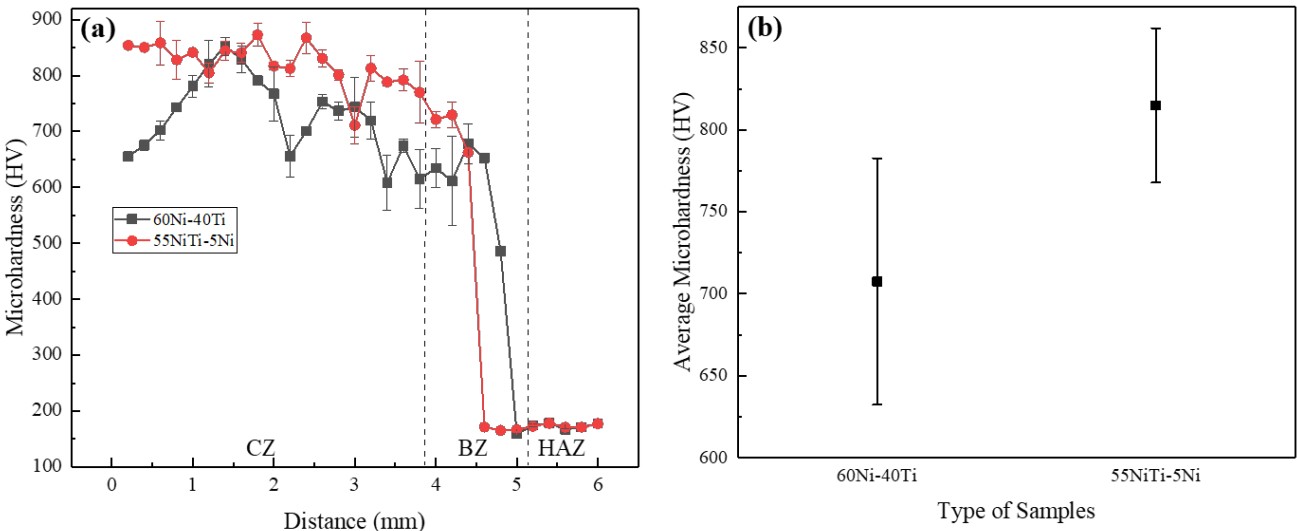

**Figure 8.** Microhardness development by depth (**a**) and average microhardness in cladding zone (**b**) of coatings.

From the top of the coatings to the substrate, the microhardness decreases slowly. This is because the top of the molten pool has a faster cooling rate, and the top structure is refined, while the bottom structure is relatively coarse due to the slow cooling rate. In addition, as the coating depth increases, the content of relatively soft substrate elements increases.

In order to further intuitively compare the hardness values of the two coatings, the average values of microhardness varied with depth of 60Ni-40Ti and 55NiTi-5Ni coatings shown in Figure 8b. The average microhardness of 55NiTi-5Ni (824.4 HV) is higher than that of 60Ni-40Ti (707.6 HV), and the standard deviation of microhardness values of 60Ni-40Ti fluctuates greatly, which is related to different microstructures [35]. 55NiTi-5Ni has a dense and uniform eutectic structure, which makes its microhardness higher than the relatively inhomogeneous and coarse structure of 60Ni-40Ti. In addition, 60Ni-40Ti has a high dilution rate and a large diffusion ratio of substrate elements, which is also a reason for its lower microhardness. The microstructure of 60Ni-40Ti is alternately composed of fine and coarse network structures, which are relatively uneven on the microscopic scale, so its microhardness fluctuates more violently.

In the bonding zone, the microhardness shows a transitional change rather than a cliff transition. As shown in Figure 9, the microhardness indentations of the two coatings are complete, and there are no microcracks around them, indicating that the bonding zone has strong toughness [36,37], which also verifies that the coatings have a good metallurgical bonding with the substrate.

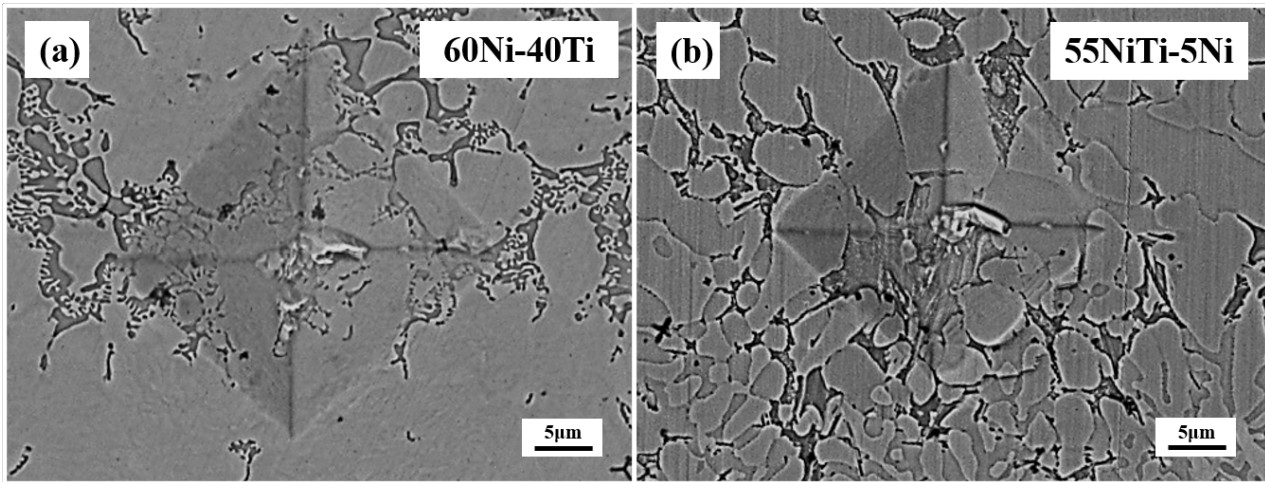

**Figure 9.** Microhardness indentation of coatings in the bonding zone: (**a**) 60Ni-40Ti; (**b**) 55NiTi-5Ni.

### 3.4. Electrochemical Behavior

Figure 10 shows the open-circuit potential curves of coatings and substrate immersed in 3.5 *wt*% NaCl solution for 2 h. As the reaction proceeds, the open-circuit potential of each sample increases slowly, which is related to the formation of passive film on the surface of the samples. The formation of a passive film on the anode surface hinders the reaction between the anode surface and the solution. As the passive film gradually dissolves, the reaction tends to balance, and the potential tends to be stable. Table 4 shows the stable open-circuit potential of the coatings and substrate. Compared with SS 316L, the open-circuit potential of the two coatings is more negative. The open-circuit potential represents the electrochemical activity of the material, which means the coatings have better electrochemical activity. The open-circuit potential of 60Ni-40Ti ($-526.1$ mV) is more negative than that of 55NiTi-5Ni ($-432.6$ mV), indicating that the electrochemical activity of 60Ni-40Ti is inferior.

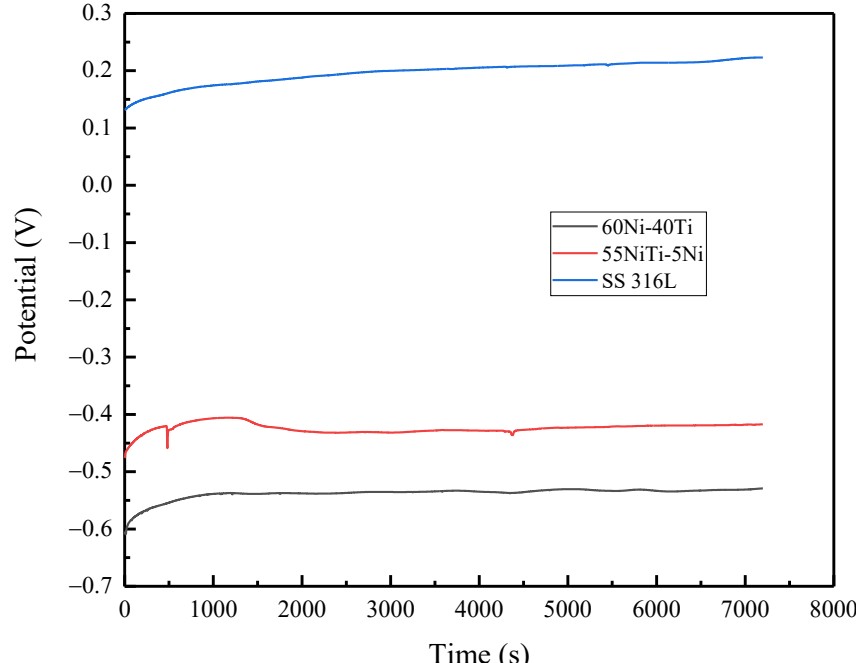

**Figure 10.** The open-circuit potential (OCP) of coatings and the substrate.

**Table 4.** The stable open-circuit potential of coatings and the substrate.

| Sample | 60Ni-40Ti | 55NiTi-5Ni | 316L |
|---|---|---|---|
| OCP (mV) | −526.1 | −432.6 | 208.1 |

Figure 11 shows the potentiodynamic polarization curves of the coatings and substrate after immersion in 3.5 *wt*.% NaCl solution for 2 h. The curves show a clear passivation platform area, indicating that a stable passivation film is formed on the surface of the material within the corresponding self-corrosion voltage range to protect the composite coating from corrosion [38]. The corrosion potential, corrosion current density and corrosion rate calculated by the Tafel fitting method are summarized in Table 5. The corrosion potential of the two coatings is higher than that of SS 316L, and the corrosion current density and corrosion rate are lower than that of SS 316L.

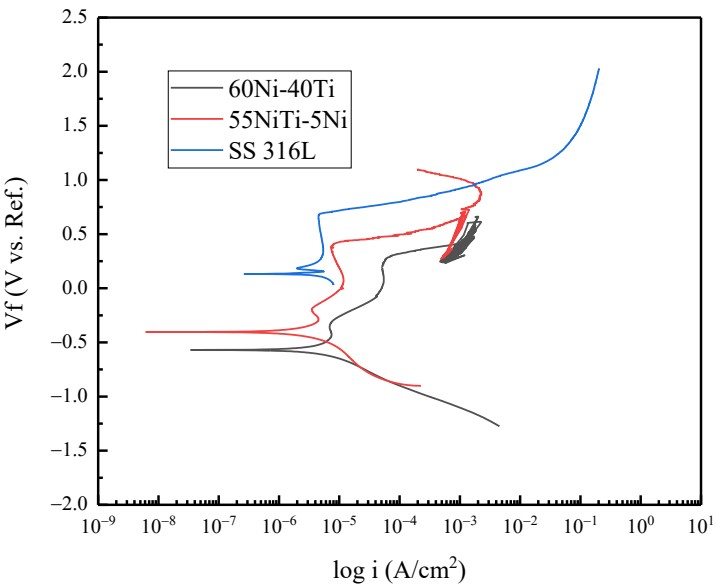

**Figure 11.** Potentiodynamic polarization curves of coatings and substrate.

**Table 5.** Tafel fitting parameters of coatings and the substrate.

| Sample | $E_0$ (mV) | $I_0$ (µA) | Corrosion Rate (mpy) |
|---|---|---|---|
| 60Ni-40Ti | −570.0 | 7.050 | 1.930 |
| 55NiTi-5Ni | −404.0 | 5.540 | 1.495 |
| SS 316L | 260.0 | 79.90 | 33.01 |

On the basis of Faraday's law, the corrosion behavior of an alloy is determined by its constituent elements with different electrode potentials, which means the higher the electrode corrosion potential, the weaker the corrosion tendency. However, the corrosion potential only indicates the corrosion tendency of the alloy and cannot replace the actual corrosion rate, which is mainly determined by the protective effect of the passive film.

The corrosion resistance of SS 316L is related to the formation of the chromium oxide film. Although iron can form an oxide film, the film is loose and not compact, which cannot prevent further corrosion of the surface. The standard electrode potential of chromium (−0.74 V vs. SHE) is lower than that of iron (−0.44 V vs. SHE). Therefore, chromium is oxidized before iron, and its oxide film is dense, which can effectively separate the sample surface from the corrosive medium. The corrosion resistance of Ni-Ti-based alloy is related to the formation of the titanium oxide film. Compared with chromium, titanium has a lower standard electrode potential of −1.63 V (vs. SHE) and has a stronger ability to form an oxide film [39]. In addition, the $TiO_2$ film has an excellent self-healing ability, which

means once the original TiO$_2$ film is destroyed, a new TiO$_2$ film can be formed immediately, which is the key reason why the TiO$_2$ film has better corrosion resistance [40–42]. In our study, compared with the substrate, although the corrosion potential of the two coatings is lower, which means they corrode faster; however, their corrosion rate and corrosion current are obviously lower (Table 4), which means that the coatings have strong corrosion resistance and improve the stainless steel corrosion resistance significantly.

Figure 12 shows the surface morphologies of the corroded coatings and substrate after the potentiodynamic polarization tests. The corrosion pits on the surface of SS 316L are larger (>100 μm), while the surfaces of the two coatings are small (<10 μm) corrosion pits, which also proves that the corrosion resistance of the coatings is significantly improved compared to SS 316L substrate.

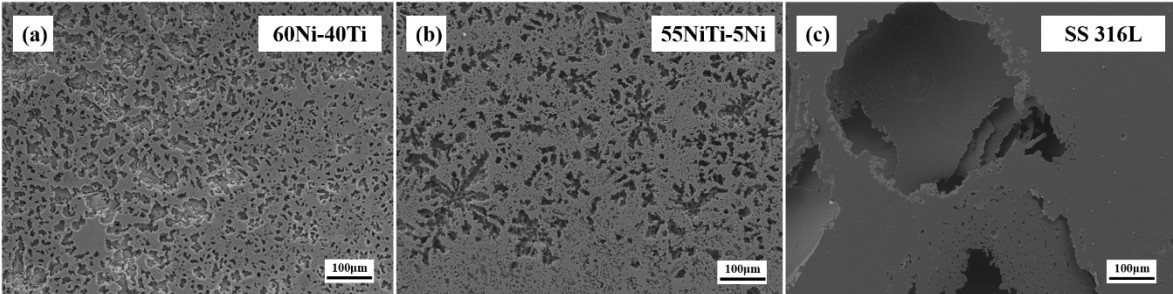

**Figure 12.** Corrosion morphology of coatings and substrate: (**a**) 60Ni-40Ti; (**b**) 55NiTi-5Ni; (**c**) SS 316L.

Table 5 also shows that the corrosion current and corrosion rate of 60Ni-40Ti are lower than that of 55NiTi-5Ni, and Figure 12 shows that the corrosion pit area of 55NiTi-5Ni is smaller, which means 55NiTi-5Ni has better corrosion resistance. To further explain the electrochemical behavior of the two coatings, electrochemical impedance spectra (EIS) tests were conducted in 3.5 *wt*% NaCl. As Nyquist plots exhibit in Figure 13, the X-axis of the diagram is the real part of the electrode impedance, which mainly reflects the impedance of the anode, and the Y-axis is the imaginary part of the impedance, which mainly reflects the capacitive reactance and inductive reactance of the anode. The Nyquist plots of the coatings are composed of a high-frequency capacitor loop, and the arc radius of the 55NiTi-5Ni is relatively large. The high-frequency capacitor circuit is controlled by the corrosion process of the working electrode as the anode. The larger the arc radius is, the more difficult the charge transfers and the better the corrosion resistance of the material.

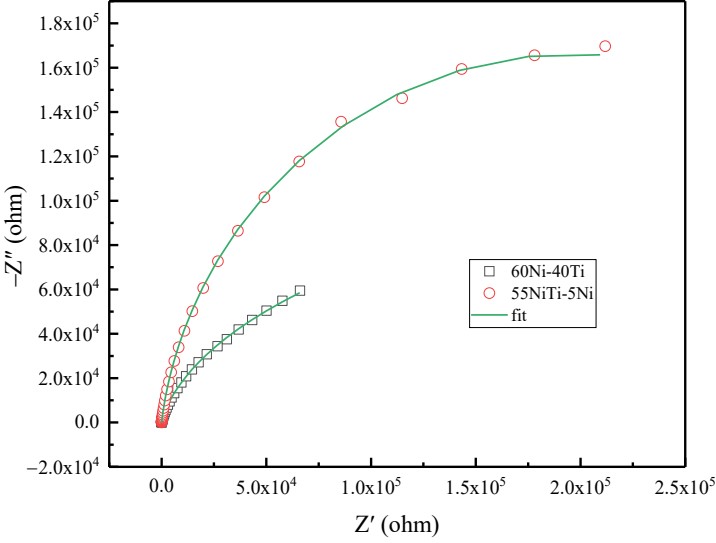

**Figure 13.** Nyquist plots of coatings.

In order to better understand the electrochemical performance of the passive film formed on the coatings, an equivalent electric circuit of electrochemical impedance spectra as shown in Figure 14 was established, and the equivalent components in the equivalent electric circuit were calculated by fitting in Gamry electrochemical software. The green solid line in Figure 13 is the electrochemical fitting curve based on the equivalent electric circuit, and Table 6 is the fitting value of each component, where the fitting deviation $X^2$ is below $10^{-4}$, indicating that the equivalent electric circuit fits well with the electrochemical corrosion system.

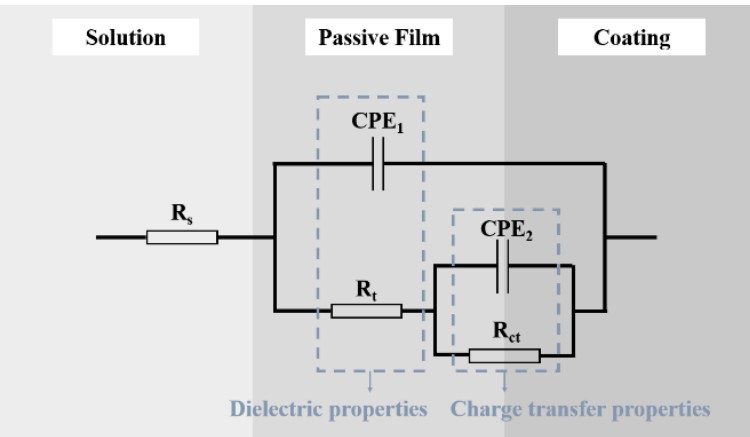

**Figure 14.** The equivalent electric circuit of coatings and a schematic representation of the passive film with the interpretation of the circuit.

**Table 6.** Nyquist plots parameters of coatings.

| Sample | $R_s$ $(\Omega \cdot cm^2)$ | $R_t$ $(\Omega \cdot cm^2)$ | $R_{ct}$ $(\Omega \cdot cm^2)$ | $CPE_1$ $(\Omega^{-1} \cdot m^2 \cdot s^n)$ | $n_1$ | $CPE_2$ $(\Omega^{-1} \cdot m^2 \cdot s^n)$ | $n_2$ | $X^2$ |
|---|---|---|---|---|---|---|---|---|
| 60Ni-40Ti | 16.02 | $9.49 \times 10^{-5}$ | $1.12 \times 10^5$ | $1.98 \times 10^{-5}$ | 0.327 | $4.30 \times 10^{-5}$ | 0.559 | $9.99 \times 10^{-5}$ |
| 55NiTi-5Ni | 16.7 | $2.216 \times 10^5$ | $1.84 \times 10^5$ | $2.751 \times 10^{-5}$ | 0.927 | $7.96 \times 10^{-6}$ | 0.891 | $3.56 \times 10^{-4}$ |

The equivalent components in an equivalent electric circuit have different meanings. $R_s$ represents the resistance of the 3.5 *wt*% NaCl solution. $R_f$ and $CPE_1$ are the resistance of the passive film and the capacitance of the double electric layer between the passive film and 3.5 *wt*% NaCl solution. The larger the $R_t$ value, the more stable the passivation film formed during the electrochemical corrosion process. The parallel connection of $R_{ct}$ and $CPE_2$ represents the charge transfer resistance and the capacitance of the electric double layer between the coating and 3.5 *wt*% NaCl solution when the penetration of ion in solution through broken passive film or defects occurred. $R_{ct}$ determines the difficulty of charge transfer. The larger the value of $R_{ct}$, the greater the resistance value generated during the electrode reaction process, and the more difficult it is to transfer the charge. $CPE_1$ and $CPE_2$ are constant phase angle elements in the electrochemical corrosion process. $Z_{CPE} = (Z_0(jw)^n)^{-1}$, where $Z_0$ is the constant of CPE, $j^2 = -1$ is the imaginary number, $\omega = 2\pi f$ is the angular frequency, and $n(-1 \leq n \leq 1)$ is the strength of the dispersion effect, and its value depends on the roughness of the passivation film on the electrode surface and the uneven distribution of corrosion current density [43]. In general, the larger the value of $n$, the denser the passive film and the more uniform the current density on the electrode surface. The results in Table 6 show that the $R_s$, $R_t$ and $R_{ct}$ of 55NiTi-5Ni are all higher than 60Ni-40Ti, and the $n$ is larger, indicating that 55NiTi-5Ni has better corrosion resistance, which is also consistent with the potentiodynamic polarization test results.

It is well known that the corrosion behavior of NiTi-based alloys, especially passivity, strongly depends on the surface conditions. According to Shen [44] et al., the passivation

of a laser-cladding coating is affected by the microstructure in the coating. The smaller tissue size provides more active sites for the nucleation and growth of the passivated film. Compared with 60Ni-40Ti, the uniform microstructure and fine eutectic microstructure of 55NiTi-5Ni are favorable to the formation of the dense oxide film.

Generally, different phases in nickel–titanium alloys form micro-cells and affect corrosion resistance [45,46]. The type and number of different nickel–titanium phases affect the electrode potential difference and the number of micro-cells, respectively. Neelakantan [47] et al. found that the interfaces of phase NiTi matrix and other nickel–titanium phases were more susceptible to pitting, and the passive film was easily destroyed. Vandenkerhove [48] et al. reported that a nickel–titanium alloy containing mainly NiTi phase showed a lower corrosion intensity. In this study, $(Ni, Fe)_3Ti$ shows a large electrode potential with $(Ni, Fe)Ti$ [6], and the proportion of $(Ni, Fe)_3Ti$ is higher in 60Ni-40Ti, which explains why the corrosion resistance of 60Ni-40Ti slightly deteriorated. In addition, due to its higher dilution rate, more elements such as Fe and Cr diffuse in the substrate. These factors lead to local corrosion or micro-galvanic effects to appear; the passivation film weakens and breaks through earlier and starts local corrosion.

## 4. Conclusions

Two kinds of 60 *wt*% Ni nickel–titanium coatings of 60Ni-40Ti and 55NiTi-5Ni, were successfully prepared on SS 316L substrate using laser cladding technology. The properties of the two coatings were significantly improved, and the 55NiTi-5Ni coating showed better performance in the microhardness test and corrosion resistance test. The results are summarized as follows:

(1). 55NiTi-5Ni coating has a lower dilution rate than 60Ni-40Ti, which means a better cladding quality.

(2). 55NiTi-5Ni coating shows a typical dendritic eutectic structure, which is composed of a dendritic and interdendritic eutectic structure, but 60Ni-40Ti tends to form a eutectic network structure composed of a coarse network structure with refined areas. XRD results show that the main phases in both coatings are $(Ni, Fe)Ti$ and $(Ni, Fe)_3Ti$, but the proportion of the $(Ni, Fe)Ti$ phase is higher in 55NiTi-5Ni along with more $(Ni, Fe)_3Ti$ in 60Ni-40Ti.

(3). The average microhardness of 60Ni-40Ti and 55NiTi-5Ni is 815.0 and 707.6 HV, respectively, which is considerably higher than substrate SS 316L, which is attributed to grain refinement and nickel–titanium particle agglomeration. Moreover, the microhardness of 55NiTi-5Ni is slightly higher than that of 60Ni-40Ti, which is related to its uniform and fine eutectic structure.

(4). Compared with SS 316L, the corrosion resistance of NiTi coatings is improved dramatically. The 55NiTi-5Ni coating shows better corrosion resistance because its uniform microstructure and fine eutectic structure facilitate the formation of the dense oxide film. In addition, its lower proportion of the $(Ni, Fe)_3Ti$ phase might induce the formation of fewer micro-cells on the surface and fewer pitting initiation points.

**Author Contributions:** Conceptualization, Z.H.; methodology, Y.F.; validation, Z.D. and Y.F.; formal analysis, Z.D., Z.H. and Y.F.; investiga-tion, Z.D.; data curation, Z.D.; writing—original draft preparation, Z.D.; writing—review and ed-iting, F.M.; project administration, Z.H.; funding acquisition, F.M. and Z.H. All authors have read and agreed to the published version of the manuscript.

**Funding:** This research received no external funding.

**Conflicts of Interest:** The authors declare no conflict of interest.

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
