# Peer review of "The Effect of Powder Composition on the Microstructure and Corrosion Resistance of Laser Cladding 60NiTi Alloy Coatings on SS 316L"

_metals, doi:10.3390/met11071104_

Round 1

Reviewer 1 Report

In this paper, 60wt.% NiTi powders were prepared using pure Ni and Ti powders and from the addition of Ni powder in 55NiTi alloy powder. After milling, the powders were used to coat the surface of 316L stainless steel by laser cladding technology to analyze the differences in microstructure and properties. The subject is interesting, with a new purpose to prepare a high-cost alloy and can aggregate value to this research field. The paper needs a grammatical revision by a professional of grammar edition. Several typos were found in the text. The introduction is quite simple, contains confusing information and undefined acronyms. What is 440C? The subject to be studied is not well presented in the introduction but contains current and adequate references (around 58% in the last five years). The methods are clearly presented and are the classical methodology for preparing and characterizing this class of materials. The results are interesting, and their analysis is adequate, based on adequate earlier findings. The conclusions are solid, supported by very interesting results. This paper can add interesting knowledge to produce NiTi alloys by powder metallurgy. I suggest its publication after revision taking into account the suggestions above presented.

Author Response

Dear Reviewer,
thank you for your positive comments on our manuscript. The manuscript has been revised according to your suggestion. You may find the responses to your comments in the attached PDF file.

If any questions, please feel free to contact me.

Zexu Du

Best wishes from Shanghai

Reviewer 2 Report

This manuscript concerns the study of the microstructure, microhardness, and corrosion resistance of laser cladding of NiTi alloy coatings.

The manuscript presents some merits and, after careful revisions, may be a candidate for publication in this journal.

Introduction:

  • The introduction should include the industrial application of NiTi alloy
  • The innovative part of this work is not clearly highlighted with respect to the current literature.
  • English must be polished. Some sentences are redundant, like lines 32-35 and 57-59.
  • In scientific English, it is better to avoid starting a sentence with And (like line 61).

Experimental methods

  • How were selected the process parameters? 
  • Please provide references for the identification of phases in XRD patterns (Figure 1)
  • Please add if the microhardness test follows a specific ASTM standard

Results and discussion:

1) Please provide references for the phases reported in the XRD patterns (Figure 2).

2) 113 typo for paragraph number.

3) For the microstructure investigations, it will be better to compare the results with the available literature in order to show similarities and differences like for lines 135-144.

4) I would recommend the authors to discuss the grain size also providing images at lower magnification (since they discuss possible grain refinement for the hardness ).

5) Regarding the microhardness, it will be interesting to compare the results with the common value of this material (also processed with other technologies).

6) in the conclusion section, it would be better to explain the positive aspects of this work before starting to describe the obtained results.

Author Response

(The authors gave the same response as above.)

Reviewer 3 Report

see the file attached, please.

Author Response

(The authors gave the same response as above.)

Round 2

Reviewer 2 Report

The authors improved the quality of the manuscript.

However, some issues are still present.

1) I cannot find references for the phases indicated in the XRD spectra in lines 96 and 137, as indicated in the answer of the authors. 

As mentioned in the previous revisions, I would recommend providing some references for the xrd spectra of the same phases. 

2) Some parts result difficult to read. 

For instance:

"The figure 4(g)-(i) show the microstructure of
195 the two coatings at a lower multiple (2000×), in which 60Ni-40Ti includes two kinds of microstructure:
196 coarse and refined. The dendritic grain size of 55NiTi-5Ni is slightly finer than that of the coarse
197 network structure in 60NiTi, while the eutectic grain size of 55NiTi-5Ni is much finer than that of the
198 refined network structure in 60NiTi. In other words, compared with 60Ni-40Ti, the overall grain size
199 of 55NiTi-5Ni is finer."

This text can be written in a more linear way.

3) there are some typos in the texts like line 62. "aser" instead of laser.

Author Response

(The authors gave the same response as above.)

Reviewer 3 Report

accepted!

Author Response

Dear Reviewer,

thank you very much for your acceptance of our manuscript (Manuscript ID: metals-1282318).

Sincerely,

Zexu Du

Best wishes from Shanghai